# Effects of Mechanical Site Preparation on Microsite Availability and Growth of Planted Black Spruce in Canadian Paludified Forests

**Mohammed Henneb [1,\*], Osvaldo Valeria [1], Nelson Thiffault [1,2]** **, Nicole J. Fenton [1] and Yves Bergeron [1]**

[1] Institut de Recherche sur les Forêts (IRF), Université du Québec en Abitibi-Témiscamingue, 445 boul. de l'Université, Rouyn-Noranda, QC J9X 5E4, Canada

[2] Natural Resources Canada, Canadian Forest Service, Canadian Wood Fibre Centre, 1055 rue du PEPS, P.O. Box 10380, Stn Sainte Foy, QC G1V 4C7, Canada

\* Correspondence: mohammed.henneb@uqat.ca

**Abstract:** Low productivity caused by paludification in some parts of the closed black spruce (*Picea mariana* (Mill.) B.S.P) dominated boreal forest threatens the provision of ecosystem services, including wood fiber production. The accumulation, over time, of organic matter in paludified soils leads to an anaerobic environment that reduces microbial activity, decelerates decomposition of organic matter, and generates nutrient-poor microsites for regeneration. Consequently, it results in significant impacts on site productivity. Considering its ability to disturb the soil, mechanical site preparation (MSP) is viewed as a potential treatment that can help restore productivity of paludified sites following harvesting. We conducted a field experiment to verify if (1) the availability of microsites conducive to reforestation varies with MSP, microtopography (slope and aspect) and initial OLT conditions; (2) the growth of planted seedlings depends on the intensity of mechanical disturbance of the organic layer, type of microsite, planting density, presence of Ericaceae, and the planting position and depth; (3) there are direct and indirect causal relationships between microsites availability after MSP, OLT, microtopography, planting quality and seedlings growth; and (4) if mechanical site preparation and microsite type exposed affect the Ericaceae cover after planting. Our results confirmed that MSP is effective in establishing conditions that permit a productive regeneration cohort on these paludified sites. To ensure successful establishment of plantations on these sites, it is necessary, however, to distinguish between those that are slightly or moderately paludified from those that are highly paludified, as treatment effectiveness of different MSP types depends on organic layer thickness. Our results also show that preference should be given to some microsite types as clay and mixed-substrate microsites for planting to ensure sufficient availability of water and nutrients for seedlings.

**Keywords:** mechanical site preparation; microsite; reforestation; productivity; competition

## 1. Introduction

Forests dominated by black spruce (*Picea mariana* (Mill.) B.S.P) occupy a large portion of the boreal biome of northeastern Canada, and are an important source of wood for the lumber, and pulp and paper industries [1]. In addition to their economic role, black spruce-dominated forests play key ecological functions, for example as a significant carbon sink [2]; however, the low productivity caused by paludification in some parts of this ecosystem threatens the provision of ecosystem services [3,4]. Paludification is a natural phenomenon characterized by an accumulation, over time, of organic layers

(from top to bottom: fibric, mesic, humic) above the mineral soil [5,6]. Consequently, paludified soils have an organic layer thickness that exceeds 40 cm, and in some cases, 100 cm [7].

On the Clay Belt of northeastern Canada, the long fire interval permits the accumulation of thick organic layers in this region [8,9] and the relatively cold climate and poorly drained soils [10] leads to an anaerobic soil environment that reduces microbial activity and decomposition of organic matter [8,9,11]. The resulting gradual accumulation of organic matter is often associated with *Sphagnum* species on the forest floor [12], leading in the long run to nutrient-poor microsites for regeneration [13,14], An abundance of such microsites contributes to reduced growth of trees, both mature and regenerating [15], with significant impact on site productivity.

Successful establishment and growth of conifer plantations on paludified sites depends on the type of microsite, the microtopography, the presence of competing species (notably ericaceous shrubs) and the quality of planting (planting position, seedling verticality, planting depth) [16–18]. The effect of planting quality on seedling growth has not been fully documented in paludified sites; this knowledge is necessary to ensure stand resilience in these ecosystems. For example, microtopography is expected to have significant effects on the availability of microsites following mechanical soil preparation (MSP), as well as on microclimate and environmental conditions at the seedling level [7]. Moreover, ericaceous shrubs are significant competitors for soil resources [19]; they can impair the successful establishment of conifers and delay the growth and survival of planted seedlings [20]. Soil disturbance through MSP, such as scarification [21], appears effective in reducing the negative effect of ericaceous competition on seedling growth. However, microsites created by scarification are quickly re-invaded by ericaceous plants [19]; therefore, the beneficial impact of the treatment can be short term. Given that thick organic layers favor the vegetative reproduction of ericaceous species [22,23], it is important to verify how MSP impacts ericaceous re-colonization of disturbed paludified sites.

The thickness of the organic layer may affect the establishment of planted seedlings, even after MSP [7,24]. MSP through light or intense scarification appears to be effective in reducing organic layer thickness and competing plant cover while creating microsites that are conducive to good rooting [25,26]. MSP has mixed effects on the availability of nutrients for regeneration, apparently depending on the treatment used and the extent of disturbance [17,27]. However, little information is available about the types of microsites created by MSP on paludified sites. Such knowledge is needed to assess the potential for silvicultural treatments to maintain or increase productivity on sites subjected to paludification and to identify microsites that should be favored during planting.

Our objectives were thus: (1) to assess how mechanical disturbances caused by three post-harvest MSP treatments (scarification with several parallel passes; plowing with two perpendicular passes, and no MSP as a control) affect microsite type availability in paludified areas; (2) to determine how the three treatments and the microsites thus created affect the growth of planted seedlings; (3) to assess how the organic layer thickness (OLT), presence of Ericaceae, microtopography and quality of planting affect the success of seedling establishment; and (4) to identify the possible relationships among MSP treatment, type of microsite exposed, and the presence of post-planting Ericaceae [18,28] on long-term forest productivity [29,30]. To these ends, we established an experimental design to test the following hypotheses: (1) the availability of microsites conducive to reforestation varies with MSP, microtopography (slope and aspect) and initial OLT conditions; (2) the growth of planted seedlings depends on the intensity of mechanical disturbance of the organic layer, type of microsite, planting density, presence of Ericaceae, and the planting position and depth [18]; (3) there are direct and indirect causal relationships between microsites availability after MSP, OLT, microtopography, planting quality and seedling growth; and (4) mechanical site preparation and microsite type exposed affect the Ericaceae cover after planting.

## 2. Materials and Methods

### 2.1. Study Site and Experimental Design

The study was located about 80 km northeast of the village of Villebois (49°06′ N, 79°08′ W), within the spruce-moss bioclimatic domain of Quebec, Canada [31] (Figure 1), more specifically in the most northerly portion of the Clay Belt (which corresponds to the distal margin of the final Cochrane surge) [32]. The clay soil is associated with extensive peatlands and topography is relatively flat. The average annual temperature is 0.1 °C and the average annual precipitation is 782 mm [33]. Black spruce and jack pine (*Pinus banksiana* Lamb.) dominate forest composition, accounting for 79% and 16%, respectively, of the forest cover, followed by trembling aspen (*Populus tremuloides* Michx), tamarack (*Larix laricina* [Du Roi] K. Koch), balsam fir (*Abies balsamea* [L.] Miller) and white birch (*Betula papyrifera* Marshall) [34]. The forest floor is covered with Sphagnum species (*Sphagnum capillifolium, Sphagnum russowii, Sphagnum angustifolium*), feather mosses (mainly *Pleurozium schreberi* [Brid.] Mitten) and shrubs (mainly Ericaceae such as *Kalmia angustifolia* L. and *Rhododendron groenlandicum* (Oeder) Kron & Judd) [4,35].

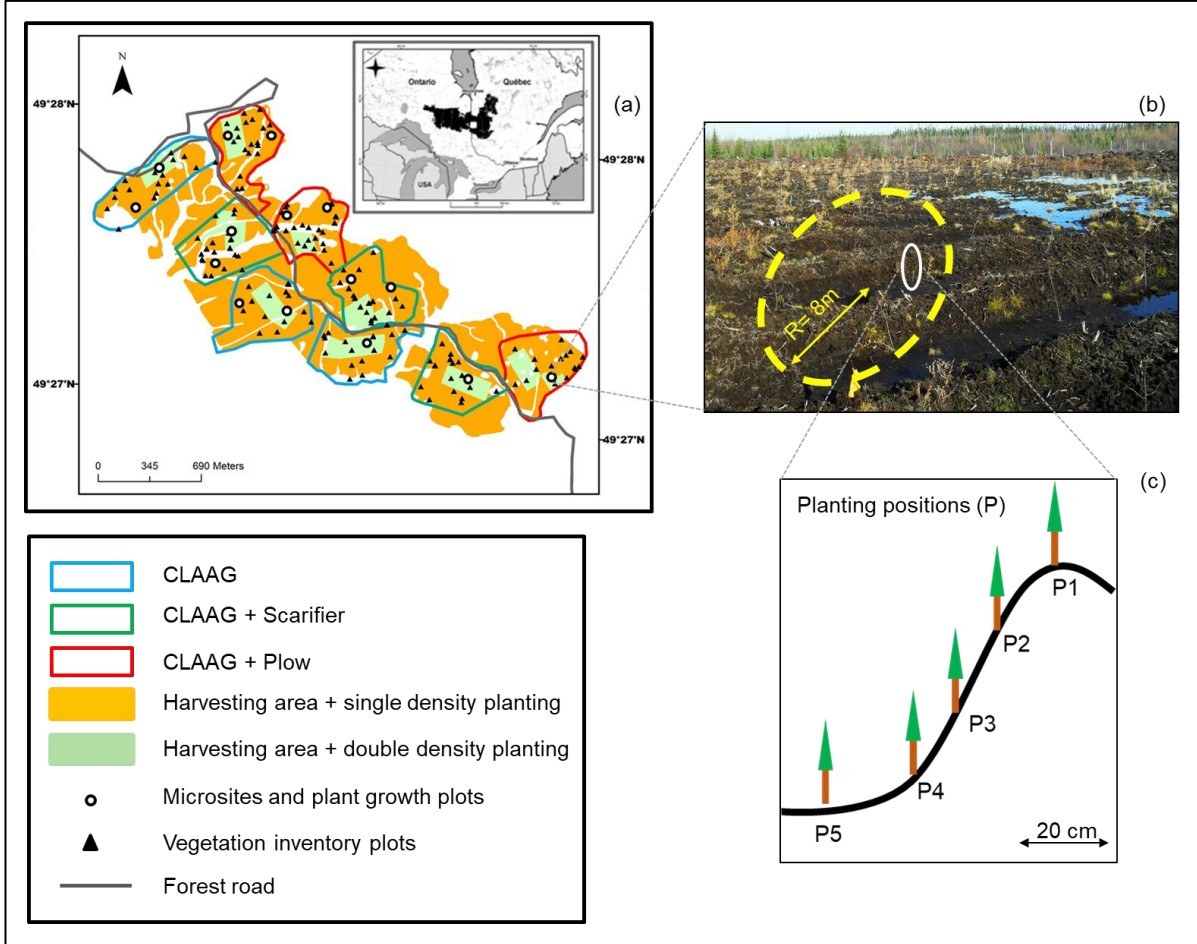

**Figure 1.** (**a**) Location of the study area (inset), distribution of the silvicultural treatments (careful logging around advance growth (CLAAG) without mechanical site preparation (MSP), CLAAG + scarifier, CLAAG + plowing), distribution of microsite and plant growth plots, and vegetation plots over the nine main cut block, and planting densities single (2200 stem per ha) and double (5000 stem per ha); (**b**) An inventory plot 8 m in radius; (**c**) Planting positions along a furrow (P1 higher to P5 lower) after mechanical site preparation.

We marked off nine cut-blocks averaging 32 ha each. The cut-blocks were harvested in the fall of 2010 using careful logging around advance growth (CLAAG, or *Coupe avec Protection de la Régénération et des Sols* (CPRS) in Quebec [36]). In the summer of 2011, each cut-block was systematically sampled for organic layer thickness at intervals of 20 m with a graduated probe, along eight parallel geo-referenced transects that were about 400 m long and 20 m apart, and oriented perpendicular to the logging trails. Microtopographical variables (slope and aspect [North, East, South, West, NE, NW, SE, SW]) were obtained from the Digital Terrain Model (DTM) (1 m resolution) derived from Lidar available data using ArcGIS software [37]. Post-harvest OLT of in the plots varied between 0 and 100 cm.

In the fall of 2011, six of the nine cut-blocks underwent mechanical site preparation. Three of the six were treated by plowing (two perpendicular passes) using a custom forest plow, and three were treated by disc trenching (parallel passes) using a T26 scarifier (Bracke Forest AB, Bräcke, Sweden). The three remaining cut-blocks served as controls, i.e., CLAAG but no mechanical preparation (Figure 1). During the summer of 2012, the sector was planted and each cut-block had two initial densities: single (2200 stems per ha) and double (5000 stems per ha) (Figure 1). All cut-blocks were planted with black spruce seedlings (initial average height = 20 cm) that had been produced in containers of 45 cells with a 110 cm$^3$ volume. During summer 2014, 15 circular sampling plots (5 per treatment), each having a radius of 8 m, were located in the cut-blocks in order to identify the availability of regeneration microsites and monitor seedling growth (Figure 1).

## 2.2. Data Collection

In summer 2014 and summer 2015 (the second and third growing seasons after planting), we measured the height (cm) and ground-level diameter (mm) of 600 black spruce seedlings in the 15 sampling plots (40 seedlings per plot) (Figure 1a,b). Sampling plots were distributed to include both classes of paludification (low to moderate, and high paludification) (Supplementary Materials). Five seedlings (two in CLAAG, two in CLAAG + plowing, and one in CLAAG + scarifier) were found dead in 2015 as a result of frost heaving. We continued to monitor the 595 remaining seedlings, characterizing microsites at the same time (Supplementary Materials). To this end, we determined (i) the degree of decomposition of the organic matter using the Von Post Scale [38]; (ii) the verticality of the seedlings (a vertical seedling is one whose inclination is within ±30° from the vertical; otherwise the seedling is deemed non-vertical [39]) and the position of seedling along the furrows formed by mechanical site preparation (Figure 1c); (iii) the depth (in cm) of planting by measuring the position of the collar with respect to ground level; iv) the existence of obstacles (stumps, rocks, etc.) near the seedlings; (v) the thickness (in cm) of the humus at the bottom of the furrows created by the scarifier or the plow, and (vi) the width (in cm) and the depth (in cm) of the furrows with respect to ground level. We also measured the distance (in cm) between the seedlings and the nearest ericaceous plant [19].

A parallel study followed the evolution in the vegetation (Ericaceae cover) in a set of 120 sampling plots (radius 11.28 m) randomly distributed across the cut-blocks (Figure 1a). Within these 400 m$^2$ plots, percent recovery of Ericaceae was measured in five 1 m$^2$ quadrats located in the north, east, south, west and center of each sampling plot.

## 2.3. Data Analysis

All analyses were conducted in the R software environment version 3.5.1 [40]. To test hypothesis (1) (availability of microsites), a non-parametric regression tree method was used (rpart, tree and mvpart packages of the R environment), in order to partition the data and identify the complex interactions among the silvicultural treatments, microtopography (slope and aspect), initial paludification conditions (post-CLAAG OLT) and the availability of microsites for seedling growth. The regression tree method, frequently used in soil science (e.g., [41,42]), works by binary splitting of the response variables into small homogeneous groups (terminal nodes) based on the numerical and categorical explanatory variables [43].

We used seedling height and diameter to calculate the relative growth rate in volume index (RGRV) [44]. The volume index (V) in $cm^3$ of each seedling was determined using the equation for the volume of a cone:

$$V = \pi \times (D/2)^2 \times (H/3) \tag{1}$$

where D is ground-level diameter (cm) and H is height (cm) of the seedlings. The relative growth rate was then calculated as:

$$RGRV = [\ln (V1) - \ln (V0)]/[t1 - t0] \tag{2}$$

where V1 and V0 are seedling volumes at time t1 (2015 growing season) and t0 (2014 growing season) [45].

To test hypothesis (2) (seedling growth), sixteen explanatory variables were incorporated in a general linear mixed model and underwent stepwise, backward-forward selection of variables and their interactions (all two-way and three-way interactions between variables) based on the Akaike Information Criterion (AIC) (stepAIC (), MASS packages of the R environment) to identify the best predictive model that explains seedling growth. The variables were assigned to five groups: (1) the "treatment" variables (scarifier, plow and CLAAG alone); (2) the "environmental conditions" variables, i.e., types of microsite exposed, microtopography (slope and aspect) and presence of competing species (planting density and distance from Ericaceae); (3) the "planting quality" variables (planting position, seedling verticality, planting depth); (4) the "initial paludification conditions" variables post-CLAAG [7]: class 1 (low to moderate paludification with post-CLAAG OLT ≤ 40 cm) and class 2 (high paludification with post-CLAAG OLT > 40 cm) [7,46]; (5) "% OLT reduction" after each treatment calculated as:

$$\% \text{ OLT reduction} = (\text{post-treatment OLT} - \text{post-CLAAG OLT})/(\text{post-CLAAG OLT}) \times 100\% \tag{3}$$

where a negative value indicates a reduction in OLT after mechanical site preparation, and a positive value indicates an increase. We used an analysis of variance (ANOVA) to evaluate the effect of the selected variables on RGRV; a Tukey's test (multicomp package in R) was used to compare treatments, the microsites and planting quality effects on RGRV. To test hypothesis (3) (causal relationships between variables), we used a path analysis (lavaan package in R) [47] to reveal the complex relationships among the explanatory variables and their effect on seedling growth and microsite availability. Also, to test hypothesis (4) (Abundance of Ericaceae), a multiple correspondence analysis (package FactoMineR) was applied to evaluate the relationships among the treatments, the types of microsite exposed and the presence of Ericaceae. Where necessary, data were transformed to follow a normal distribution, using α = 0.05 as the significance level.

## 3. Results

### 3.1. Availability of Microsites

We identified five main types of microsite (Figure 2): clay, organic-clay mixture, clay-humic mixture, fibric and humic [6,48]. The regression tree analysis (Figure 3) shows that the availability of these microsites varied with the treatment and the post-CLAAG OLT class (≤40 cm vs. >40 cm). The "treatment" variable splits along a left branch (CLAAG) and a right branch (plow-scarifier), leading to two significantly different daughter nodes. The post-CLAAG OLT node then splits into two significantly different terminal nodes: where the OLT was less than 40 cm, the distribution of microsite types varied significantly with the treatment ($p \leq 0.05$). CLAAG treatment without MSP resulted in 70% fibric microsites, followed by 20% humic microsites, while organic-clay microsites failed to exceed 10%, and clay and clay-humic microsites were hardly exposed at all (<1%). With the scarifier treatment, about 30% of the microsites exposed were clay, followed by clay-humic (about 25%) and organic-clay (about 20%). Fibric and humic microsites had the lowest percentages (about 15% and 10% respectively)

on scarified plots. As for the plow, it exposed more humic microsites (40%) than other types, followed by clay-humic (20%), clay (~15%), organic-clay (~15%) and fibric (10%).

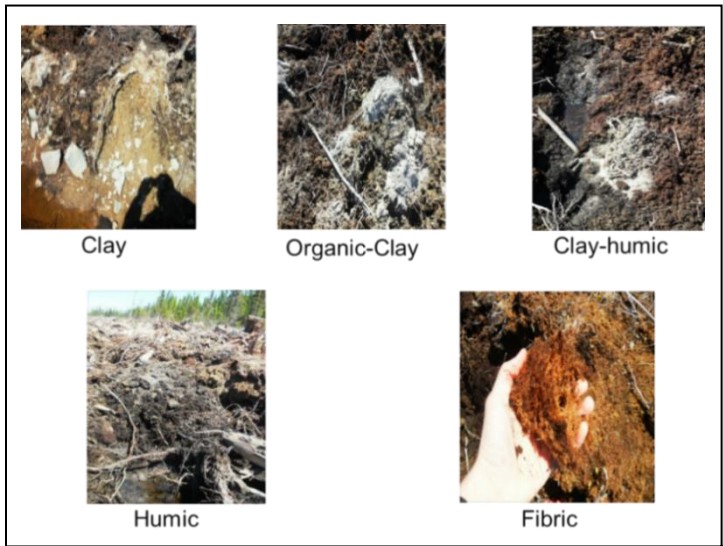

**Figure 2.** Main types of microsites found at the study site.

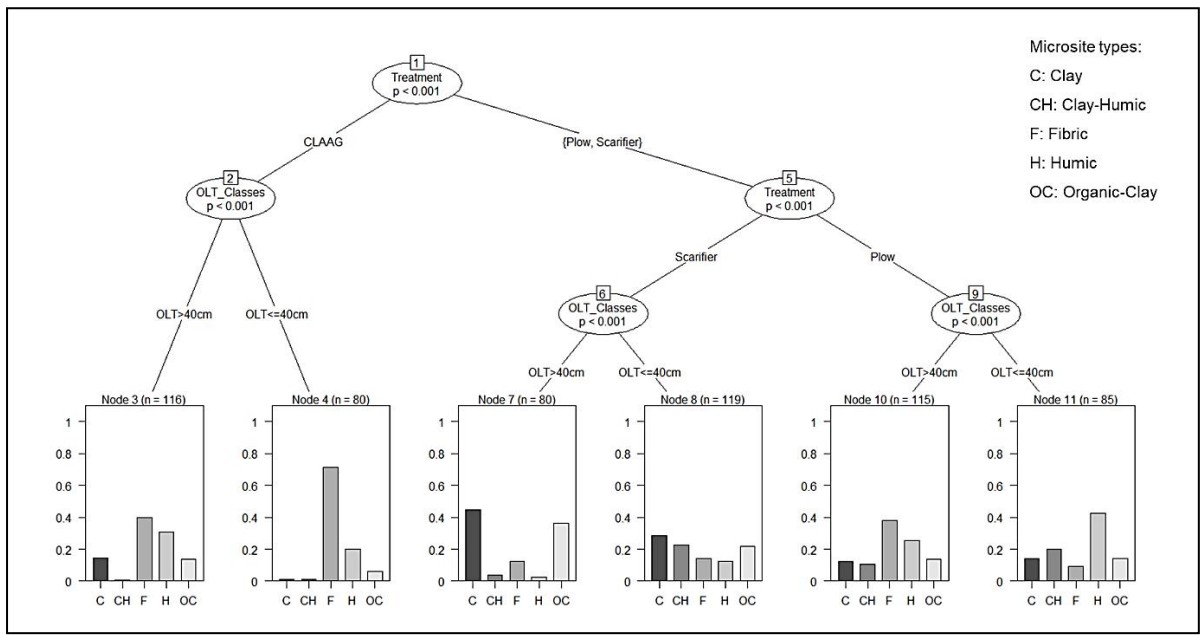

**Figure 3.** Percent microsites exposed by treatment and by post-CLAAG organic layer thickness (low to moderate OLT ≤ 40 cm and high OLT > 40 cm).

Where the OLT was greater than 40 cm, CLAAG barely exposed clay-humic microsites (<1%), but resulted in more fibric microsites (40%), followed by organic-clay (30%), clay-humic (15%) and clay (~15%). The scarifier exposed more microsites that were clay (~45%) or organic-clay (~35%) than the other types; next came fibric (~15%), humic (~5%) and clay-humic (~5%). Finally, the plow exposed more fibric microsites (~40%), followed by humic (25%), organic-clay (~15%), clay (~10%) and clay-humic (~10%).

### 3.2. Seedling Growth

Overall, the stepwise selection and ANOVA results (Table 1) showed that seedling growth was significantly influenced by silvicultural treatments, planting position, microtopography (aspect), microsite type and planting position interaction, microsite type and planting depth interaction, percent reduction in OLT and OLT post-CLAAG interaction, planting depth and seedling verticality and the interaction among silvicultural treatments, post-CLAAG OLT and percent reduction in OLT, post-CLAAG OLT, percent reduction in the OLT and microtopography (Table 1). These variables and interactions were selected as part of the best mixed model ($R^2$ = 0.497, AIC = −136.859, the worst model had an AIC = −43.965) that had more effect on seedling growth (Table 1). "Planting density" and "distance from Ericaceae" were not selected because of their weak influence on growth.

**Table 1.** Listing of variables and interactions selected by stepwise (backward-forward selections) composing the best mixed model. Summary of ANOVA results for the effect of selected variables and interactions on seedling relative growth rate in volume index (RGRV). The mixed effect model explained 49.7% of the variation in RGRV.

| Selected Variables and Variables Interactions | Df | Sum Sq | Mean Sq | *F*-Value | *p*-Value (>F) |
|---|---|---|---|---|---|
| Treatment | 2 | 4.7219 | 2.3610 | 4.5835 | 0.013 |
| Planting position | 4 | 4.9752 | 1.2438 | 2.4147 | 0.048 |
| Aspect | 6 | 6.7586 | 1.1264 | 2.1868 | 0.043 |
| Microsite type × Planting position | 16 | 15.3619 | 0.9601 | 1.8640 | 0.021 |
| Microsite type × Planting depth | 4 | 5.0100 | 1.2525 | 2.4316 | 0.047 |
| OLT reduction × OLT post-CLAAG | 1 | 3.6261 | 3.6261 | 7.0397 | 0.008 |
| Planting depth × Seedling verticality | 1 | 2.1480 | 2.1480 | 4.1701 | 0.042 |
| Treatment × OLT post-CLAAG × OLT reduction | 4 | 15.4200 | 3.8550 | 7.4840 | <0.001 |

OLT: organic layer thickness; CLAAG: careful logging around advance growth. Significance at $p \leq 0.05$.

Where post-CLAAG OLT was low to moderate (≤40 cm), CLAAG without MSP yielded lower RGRV than the other two treatments, only if the percent reduction in OLT was under −20%. Above that threshold, growth in the CLAAG treatment decreased gradually as OLT increased (Figure 4). With the plow, seedling growth increased as the percent reduction in OLT increased (i.e., lower OLT). The opposite was observed with the scarifier: seedling growth increased as the percent reduction in OLT decreased (i.e., organic matter accumulation). Nevertheless, seedling growth was better with the plow than with the scarifier when OLT was reduced.

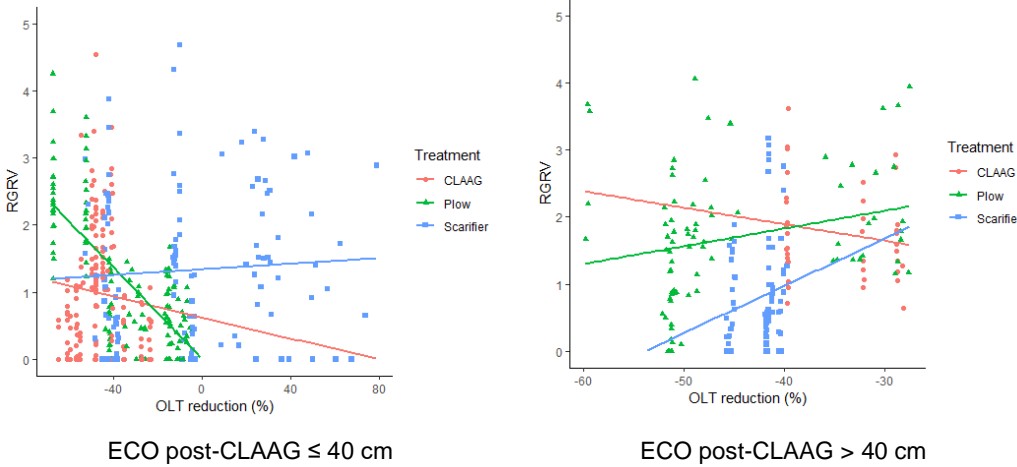

**Figure 4.** Effect of silvicultural treatments on the relative growth rate in volume index (RGRV) of the seedlings, by post-CLAAG OLT and percent reduction in OLT ($p < 0.001$). A negative value of the variable "OLT reduction" means that there was a decrease in OLT; a positive value means an increase in OLT. OLT: organic layer thickness; CLAAG: careful logging around advance growth.

Where post-CLAAG OLT was high (>40 cm) (Figure 4), seedling growth was better overall with the plow and scarifier than CLAAG treatment without MSP. Indeed, with the plow and scarifier, seedling growth increased gradually as the percent reduction in OLT decreased. The opposite was observed for CLAAG treatment without MSP.

The effect of microsite type on growth was significant only in combination with the planting position and depth. RGRV was generally better on clay microsites, especially at planting positions 1, 4 and 5 (Figure 5a). Linear regression (Figure 5b) showed that when the seedling collar was at ground level (depth = 0 cm), growth was better on clay, organic-clay and fibric microsites. When the collar was above ground level, growth was better on clay microsites. As the collar approached 10 cm below ground level, clay microsites showed the lowest growth response.

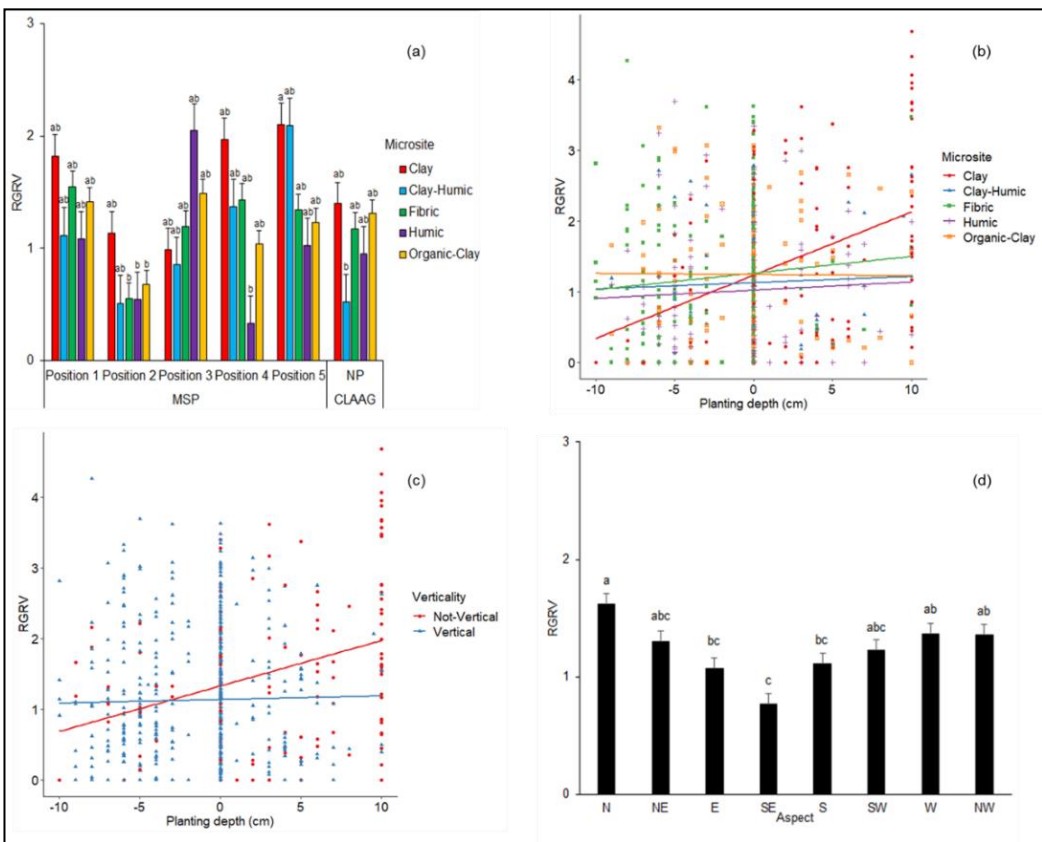

**Figure 5.** (**a**) Differences in relative growth rate in volume index (RGRV) by microsite type and planting position (see Figure 1c for a description of the planting positions). NP (No-Positions) indicates no planting position due to lack of furrow. (**b**) Effect of microsite type on the RGRV of seedlings by planting depth ($p = 0.047$). (**c**) Significant linear relationship ($p = 0.042$) among RGRV, seedling verticality and planting depth. (**d**) Differences in RGRV by aspect. Bars topped by the same letter are not statistically different ($p \geq 0.05$). CLAAG: careful logging around advance growth. 3.3. Path analysis and correlations among variables influencing growth.

ANOVA results revealed a significant effect of seedling verticality on RGRV. The effect varied with planting depth for non-vertical seedlings (Table 1). When planting depth was more than 3 cm below ground level, growth was better with non-vertical seedlings. Within vertically planted seedlings, growth was similar regardless of planting depth (Figure 5c). Finally, aspect had a significant effect on growth, which was better in N, NW and W orientations than with SE (Figure 5d).

### 3.3. Path Analysis and Correlations among Variables Influencing Growth

The path analysis (Figure 6) showed that the direct correlations between seedling growth and the following variables—post-CLAAG OLT, % OLT reduction, planting position and microsite type—did not appear to be significant. This was true for both the scarified and plowed conditions.

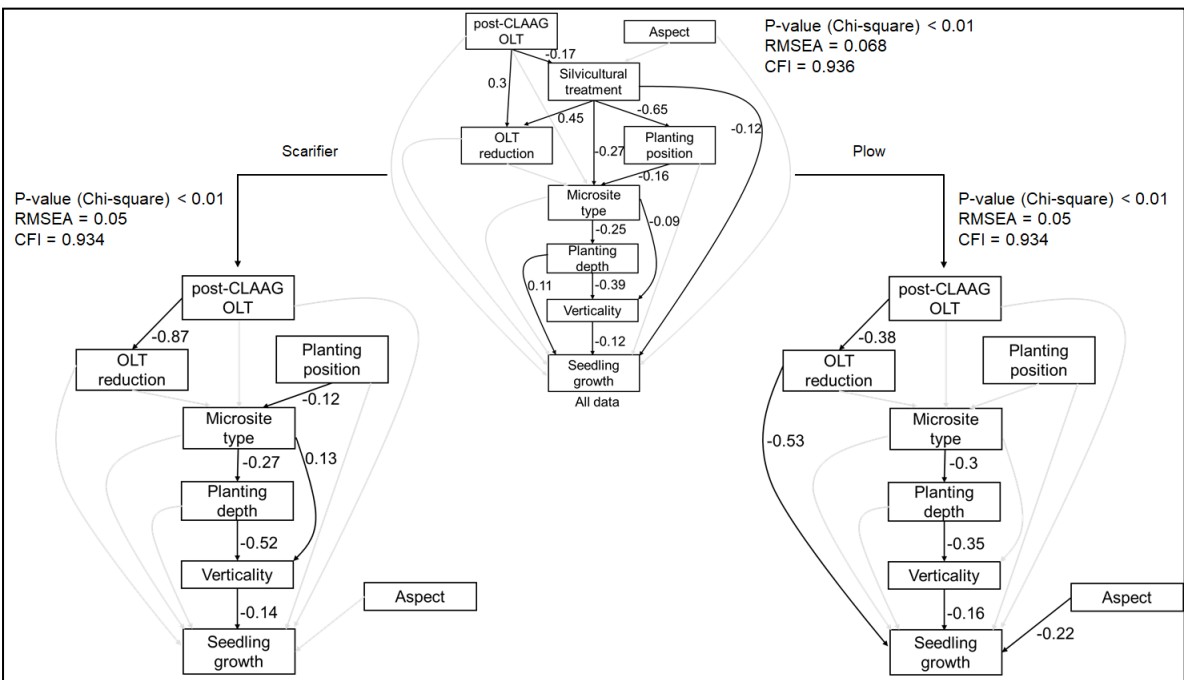

**Figure 6.** Path analysis summarizing the direct and indirect correlations influencing seedling growth, as expressed by their relative growth rate in volume index. The analysis was performed for all of the data combined and for the scarifier (**left**) and plow (**right**) separately. The variables included account for 50% of the variability. Darker arrows indicate significant correlations. The number beside each arrow is the coefficient representing the influence of each correlation. The parameters used to fit the models were the Root Mean Square Error of Approximation (RMSEA) and the Comparative Fit Index (CFI). OLT: organic layer thickness; CLAAG: careful logging around advance growth.

The effect of post-CLAAG OLT on % OLT reduction was greater in plots treated with the scarifier than in those treated with the plow (Figure 6). Treatment directly and significantly influenced OLT reduction, planting position, microsite type and seedling growth. We observed direct, significant correlations between planting position and types of microsites exposed. At locations treated with the scarifier, microsite type was linked with planting depth and seedling verticality. The path analysis also revealed a direct, significant link between planting depth and seedling verticality (Figure 6). The coefficients for these correlations were higher under the conditions created by the scarifier than under those created by the plow. Lastly, we noted a direct correlation between seedling growth, planting depth and seedling verticality. The direct influence of aspect on seedling growth was significant only in plots treated with the plow.

### 3.4. Post-Planting Ericaceae Cover

The first two axes of the multiple correspondence analysis (Figure 7) explained 36.9% of the variability in the data. *Vaccinium* species were closely associated with conditions created by the plow and humic microsites, but showed little association with clay microsites. *Rhododendron* was associated more with CLAAG without MSP, and with fibric microsites; they were scarce on organic-clay and clay-humic microsites. *Kalmia* was associated with the MSP-treated plots, in particular those that had been scarified, as well as with organic-clay and clay-humic microsites. *Kalmia* was less associated with

CLAAG without MSP and with fibric microsites. Lastly, none of the Ericaceae species were closely associated with the clay microsites, a large proportion of which were exposed by the scarifier (Figure 7).

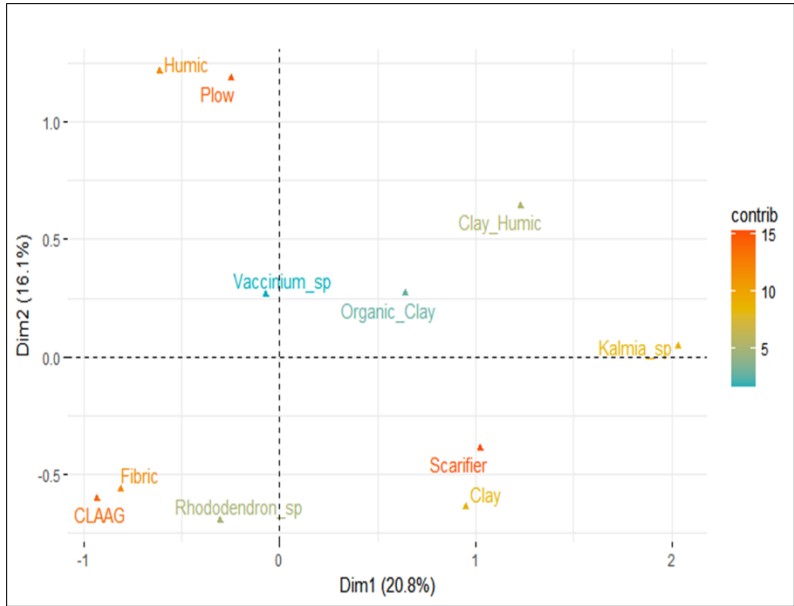

**Figure 7.** Multiple correspondence analysis summarizing the relationships among Ericaceae cover, silvicultural treatments, and types of microsites exposed. The color gradient represents the contribution of each variable on the two explanatory axes. CLAAG: careful logging around advance growth.

## 4. Discussion

### 4.1. Availability of Microsites

In general, the plow and the scarifier exposed more clay microsites, mixed (clay-humic, organic-mineral) microsites and nutrient-rich organic (humic) microsites [14], compared with the control treatment (CLAAG with no MSP), independently of OLT. The soil disturbance resulting from MSP improves survival and growth, increases nitrogen availability, and improves tree nutrition while reducing inter-species competition [24,49]. MSP also improves soil temperature, moisture, and fertility [25,26,50] and makes work easier for planters, because it relocates or eliminates part of the woody debris left by logging operations [51,52]. In contrast, the soil disturbance caused by CLAAG without MSP is limited to the skidding trails (25% of the area of the cutovers; [53]) as a way of protecting soils in accordance with the Forestry Act in Quebec. Consequently, exposure of microsites favourable to seedling growth occurs only on harvesting and skid trails, while most of the cutover remains intact.

Post-CLAAG OLT directly influenced the relative effectiveness of the MSP treatments in exposing microsites. Where post-CLAAG OLT was low to moderate (≤40 cm), the scarifier exposed more clay and mixed microsites than the plow, but the plow exposed more humic microsites (>40%) than the scarifier. Humic microsites represent an important nutrient reserve (especially N) and support nutrition and growth of plantations over the medium and long term [54]. Humus also helps to increase the soil's cation exchange capacity and regulates oxidation-reduction, thus improving availability of oxygen to the roots [55].

Where post-CLAAG OLT exceeded 40 cm, the scarifier was more effective than the plow in exposing microsites conducive to growth. The plow exposed more fibric microsites with a low nutrient content; its effectiveness was limited by the greater OLT. In contrast, the scarifier severely disturbed the thick organic layer, exposing more clay microsites (about 45%) and mixed organic-clay microsites (about 35%). Our results support those of earlier studies that demonstrated how scarifiers can increase the productivity of black spruce plantations in paludified environments [8,56,57]. The discs of the T26

scarifier are 1.35 m in diameter; they thus can reach the deep mineral horizons and mix them with the organic material.

### 4.2. Seedling Growth

Seedling growth depended on several variables and the interactions among them. In general, and as reported elsewhere [56,58], CLAAG with MSP resulted in greater growth than CLAAG alone. On sites where post-CLAAG OLT was low to moderate, seedling growth was better with plowing than with the two other treatments. However, growth diminished as the percent reduction in OLT approached 0. At sites where the reductions in OLT were large, microsites favourable to seedling growth—in particular, humic, clay, and mixed microsites—were exposed by the plow. But at sites where the reductions in OLT were smaller (i.e., where the organic layer was less disturbed), the number of microsites favourable to seedling growth that were exposed was small, because of the plow's ineffectiveness in disturbing the thick organic layer [16,24,59].

In plots treated with the scarifier, seedling growth increased gradually despite OLT accumulation. Indeed, the mounds built up on either side of the furrows by the scarifier exposes many mineral and organic-mineral microsites [7] that may favour seedling growth in the short term [24]. Generally, such planting conditions are not recommended on non-paludified sites, because of their instability and the high risk of drying out [25,26], which could affect seedling growth negatively in the long term [17,60,61].

We found that seedling growth was better on clay microsites than on other types of microsites in almost all planting positions. At this early stage of growth, access to light and water is more critical than access to other resources. While light levels are not an issue for regeneration on recently cut areas in paludified forests, access to water is better on clay microsites that are exposed at the surface (disturbed clay), since these have a high water-retention capacity [62,63] compared to other types of microsites. However, planting in bare, undisturbed clay soil entails a high risk of root asphyxiation caused by the stagnation of the water on the surface, especially in depressions [9,64–66]. We also observed that seedlings planted with the collar below ground level (planting depth < 0 cm) showed poor growth on clay microsites but better growth on organic-clay microsites. In the presence of organic material or on organic-mineral microsites, deep planting provides better access to water at greater depths and stimulates the expansion of the initial roots and growth of adventitious roots [14,67,68].

Seedlings that were planted vertically grew steadily, regardless of planting depth, whereas among the seedlings that were not planted vertically, growth increased gradually as planting depth decreased. Below the planting depth of −3 cm, a common practice in eastern Canada, the growth of the vertical seedlings was greater than the non-vertical seedlings. The advantage of the non-vertical seedlings above this threshold was probably the result of the growth of adventitious roots in contact with the moist soil [69,70].

Seedlings planted on northern slopes generally grew better than those planted on southern slopes. The reason, we believe, is that northern slopes are less exposed to solar radiation and so remain wetter than southern slopes [71,72]. However, the influence of aspect on seedling growth varies from one site to another and depends on several factors, notably site topography, OLT, and the degree of disturbance of the organic layer [7,71].

### 4.3. Relationships among the Variables That Influence Seedling Growth

The path analysis showed that on paludified sites, the effectiveness of silvicultural treatments was significantly correlated with post-CLAAG OLT; this variable determines the direct [7] and the indirect influence of treatments on other environmental variables and, ultimately, on seedling growth [25,50,52]. The characteristics of the equipment and the penetration depth of the discs probably explain the observed differences between the MSP methods that we tested (Figure 6; [25,50]). The path analysis also revealed close relationships between the planting depth and the verticality of the seedlings, which supports the importance of planting quality. Lastly, our results show how planting location, with

regard to aspect at the microtopographic scale, can be decisive for seedlings when a site is mechanically prepared with a plow [7,59].

*4.4. Abundance of Ericaceae*

The presence and distribution of Ericaceae after planting are highly correlated with post-disturbance environmental conditions, and with site fertility in particular [73–75]. These relationships were clearly present on the experimental site; we found that *Vaccinium* was more abundant on humus-rich plowed sites; *Rhododendron* was more abundant in the control plots, characterized by fibric organic material; and *Kalmia* was closely associated with microsites having high proportions of organic-clay and clay-humic mixtures. Ericaceae were less abundant on microsites with a high clay content, which are less fertile than humic or mixed sites. Also, and contrary to what we had hypothesized, our results did not show any significant relationship between the abundance of Ericaceae and the short-term growth of the seedlings (three growing seasons). We conclude that scarification and plowing reduced the Ericaceae cover sufficiently to limit their direct and indirect interference with planted seedlings [18,28].

## 5. Conclusions and Implications for Forest Management

Although the selected model explained a significant portion of the seedling growth variability (49.7%), other factors that we have not studied are significantly influential, as 50% of the variability remains unexplained. Nevertheless, our study confirms that the use of MSP to disturb paludified soils is effective in establishing a productive regeneration cohort in eastern Canada [9,17,18]. MSP with a plow provided the best growth in areas with low to moderate post-CLAAG OLT (≤40 cm). However, the scarifier performed better in areas with post-CLAAG OLT greater than 40 cm. To ensure successful establishment of plantations on these sites, it is therefore essential to distinguish between those that are slightly or moderately paludified and those that are highly paludified. Doing so will make it possible to choose the right MSP treatments and expose more microsites that are conducive to seedling establishment. MSP also enables adequate control over Ericaceae in the short term; reinvasion of the microsites over the medium and long terms remains to be documented. During planting operations, preference should be given to clay and mixed (organic-clay and clay-humic) microsites so as to ensure sufficient availability of water and nutrients. On clay microsites, seedlings should be planted fairly shallow, so as to stimulate the appearance of adventitious roots near the surface and thus give the seedlings better access to the resources (water and nutrients) available in the soil.

**Supplementary Materials:** The following are available online at http://www.mdpi.com/1999-4907/10/8/670/s1, Table S1: Number of monitored microsites and seedlings according to paludification classes.

**Author Contributions:** Conceptualization, M.H., O.V., N.T. and N.J.F.; methodology and formal analysis, M.H.; supervision, O.V. and N.T.; writing—original draft preparation, M.H.; writing—review and editing, M.H., O.V., N.T., N.J.F., and Y.B.

**Funding:** This project was made possible with funding provided by NSERC (Natural Sciences and Engineering Research Council of Canada), the NSERC-UQAT-UQAM Chair in Sustainable Forest Management, and Tembec Inc, in collaboration with the Ministère des Forêts, de la Faune et des Parcs du Québec (MFFPQ), through both its regional office and the Direction de la recherche forestière (former employer of N.T.).

**Acknowledgments:** We thank Tembec Inc and the Direction de la recherche forestière (MFFPQ) for their collaboration; the Laboratoire de chimie organique et inorganique of MFFPQ for soil analyses; and Julie Arseneault (UQAT) for her technical assistance.

**Conflicts of Interest:** The authors declare no conflict of interest.

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
