# Peer review of "Effects of Mechanical Site Preparation on Microsite Availability and Growth of Planted Black Spruce in Canadian Paludified Forests"

_forests, doi:10.3390/f10080670_

Round 1

Reviewer 1 Report

The response in the cover letter from authors does nothing to reduce my concern over inadequate sample size. In my decades of forest research, I have long realized the scope of variation encountered in field settings. It was good to learn the plots were distributed in all "divisions" of the research, but that does not mean the range of conditions or seedling response was sampled. It only states that there was a sample taken. This might (or might not) be an adequate sample in a greenhouse study, but for a treated area of the size in this study and with this many seedlings, I consider the sample to be inadequate.

Reviewer 2 Report

Greatly improved after the revisions!

This manuscript is a resubmission of an earlier submission. The following is a list of the peer review reports and author responses from that submission.

Round 1

Reviewer 1 Report

Paper: Effects of mechanical site preparation on microsite availability and on growth of planted black spruce in Canadian paludified forests

General Comments on Introduction

The Introduction is way to short! It does not do an adequate job of introducing the topics that you discuss in the paper or create a strong enough theoretical basis for the work.

Specific Comments on Methods Section and beyond.

Line 137: You mention “complex interactions among the silvicultural treatments” and appropriately use a non-parametric regression tree to assess this. Then you simplify your data by doing a model selection. Why? Keep the complexity as you state and then use appropriate models to capture this. Why revert to parametric tests when you don’t need to?

Line 150: You mention “Sixteen explanatory variables were incorporated in a general linear mixed model and underwent stepwise, backward and forward selection” What was the criteria for inclusion/exclusion of explanatory variables? Did you use AIC? You still have lots of variables listed in your final model that are non-significant. I don’t understand this if you went through a model selection criterion….

Line 200: How do you know that the result was significant? (Some factors and interaction were significant, but the majority were not. You also state an “R2 = 0.497”, this only gives you 29% of the standard deviation explained by your model. Not the best model… Maybe more reason to evaluate using non-parametric measures.

Line 226: You show lots of interactions in the Anova Table but, you did not mention these in your methods section.

I stopped reviewing after line 239 due to major issues that I have with your methods and results….

Reviewer 2 Report

The study presented in this paper  contributes to the existing silvicultural knowledge on black spruce forest reforestation in Canada. The obtained results will be of value to forestry practitioners in making decisions concerning the optimal silvicultural treatments (choice of the site preparation method, planting position and depth, seedlings densities),  to maintain or increase productivity on paludified sites. In my opinion, the study is on a topic of  relevance and interest to the readers of the journal. This investigation addresses an important issue, for which useful current research results are lacking.

The paper is original and well written, clear and easy to read. Materials and methods are adequate and sufficiently described. Data and results are presented appropriately. The discussion is well written and the conclusions clearly summarize the obtained results.

Reviewer 3 Report

Overall, the manuscript is well written and while some minor editing would improve the presentation, those items are TOTALLY secondary to my primary concern which is the validity of the analysis and results which are based on such a low sample intensity.

You report a total of 278 hectares involved in the study ( 9 blocks of 32 ha per block). That provides a total of 2,780,000 sq. meters in the study blocks. Of that total, you sampled only 15 plots (l. 106-107) which were @ 200 sq. meters per plot for a total of @3000 sq. meters for the study. Unless my math is incorrect, this is a sample of only 0.108%. Is there something missing in your description or perhaps I am misreading "15 circular sampling plots ( 5 per MSP treatment)". Why have 9 blocks in the study if you only used 15 sample plots?

 Your plots were "randomly located"  (L.107) , but the reader has no knowledge of how many of the 5 plots were in a "MSP-treated" area vs. "harvested only" within the treated blocks. Perhaps the scarifier covers 100% of the land area, but I do not think the plow would.

Likewise, your sample of the seedlings is very low. The total seedlings planted range from 611,600 (low density) to 1,390,000 (high density). You report sampling a total of 600 ( 40/plot in 15 plots) which is a possible sample intensity of 0.043% to 0.098%, depending on planting density. Forty seedlings would be almost all in the low planting density, but how were they selected if in a high-density area which would have had @ 100 seedlings in a plot?

A population must be extremely homogeneous for sample intensity of <1% to be reliable. Perhaps I have misread your presentation, but it appears that a sample of 15 plots with 600 seedlings from 278 ha and hundreds of thousands of seedlings is very low. If your sample intensity was greater, please adjust your presentation to reflect such and include some description of why you consider your sample scheme to be valid.